# MULTIL-LEVEL MULTIMODAL ALIGNMENT WITH KNOWLEDGE-GUIDED INSTANCE-WISE DISCRIMINATION

## ABSTRACT

In multimodal alignment, meta-alignment and multi-level alignment play important roles. However, it is challenging to integrate meta-alignment into a multi-level multimodal alignment framework involving the operation on both reducible substances (e.g., molecules and spectrum) and irreducible elements (e.g., atoms and spectral peaks). It not only inherits the challenges from meta-alignment (e.g., heterogeneity, loss of nuance, interference, and conflicting similarities) but also introduces new challenges: navigating the interactions among reducible substances and irreducible elements and recognizing objects at each level. Many existing alignment methods suffer from inaccurate component relation estimation and potential bias, as they hold manual definitions of pair closeness. In response, we introduce Multi-level Multimodal Alignment with Knowledge-Guided Instance-Wise Discrimination (K-M3AID), an innovative approach that utilizes continuous knowledge variables with inherent natural ordering for meta-alignment. K-M3AID effectively addresses these challenges by promoting both reliable distance learning and unbiased alignment within the context of cross-modality alignment for multi-level structures. Extensive empirical studies conducted on complex molecular structures underscore the substantial efficacy of K-M3AID. It significantly improves matching accuracy while augmenting multi-level alignment capabilities. This novel approach holds great promise for advancing alignment techniques across diverse molecular contexts, offering a more robust foundation for ongoing research in chemical analysis and beyond.

## 1 INTRODUCTION

Multimodal alignment (MMA), as a critical aspect of multimodal deep learning, aims at establishing connections between contextually related information across heterogeneous modalities (such as text, images, audio, video, sensor data, etc) (Liang et al. (2023); Jabeen et al. (2023)). Its subject may take the form of either a **reducible substance (RS)** or an **irreducible element (IE)** within a reducible substance. RS-MMA signifies a form of high-level alignment, exemplified by semantic alignment, which enables models to extract and understand the rich semantics and meanings across different modalities (Rocco et al. (2018); Wu et al. (2022); Yang et al. (2023); Liang et al. (2023)). IE-MMA, in conjunction with meta-learning(Vilalta & Drissi (2002); Vanschoren (2018); Nichol et al. (2018)), converges into multimodal meta-alignment (IE-Meta-MMA), which carries great potential for cognitive processing, generalization, and the remarkable capacity to execute zero-shot tasks (Ma et al. (2022)). While multi-level MMA (MLMMA) has been demonstrated for visual-textual alignment (Hu et al. (2019); Khan et al. (2022)), these frameworks remain limited to the only combination of multi-level RS-MMA, not involving IE-Meta-MMA. The potential integration of RS-MMA and IE-Meta-MMA can result in a synthesis of the advantages and benefits offered by both approaches, creating a unique paradigm of MLMMA.

Does the introduction of IE-Meta-MMA into MLMMA pose a significant challenge? The incorporation of IE-Meta-MMA into the MLMMA framework will not only inherit substantial challenges from Meta-MMA, such as notable data heterogeneity, limited data annotation and labeling, loss of nuance, interference, conflicting similarities, generality and transferability, but also introduce new challenges: the dependence between RS-MMA and IE-Meta-MMA. A successful MLMMA model

must attain the following capabilities: a) preform effective representation learning for multimodal information with varied data formats, scales, and noise levels; b) proceed dynamic communication between RS-MMA and IE-Meta-MMA that accommodate the dependence and interaction among different level alignments; c) decipher complex relationships between RSs and IEs within dynamic environments. MLMMA calls for algorithmic sophistication, interdisciplinary collaboration, and a holistic understanding of data interplay.

MMA has also emerged as a catalyst to revolutionize the field of chemistry, particularly establishing the correspondence between molecules and their functionalities (Finlayson et al. (2020)) or expressions through a variety of spectroscopes (Yang et al. (2021)). In view of that molecules come into existence by the union of atoms, these molecular-level interplays are categorized into RS-MMA. Apparently, these interplays don't offer profound atomic-level insights. A solid understanding of atomic characteristics and functions with specific local contexts can enhance our understanding of molecular-level phenomena. And these meta-knowledge can be generalized and applied to diverse situations with a high degree of precision, even in zero-shot scenarios. Potentially, it could aid in solving isomer recognition, one of the most challenging tasks in chemistry (Bifulco et al. (2007); Duddeck & Díaz Gómez (2009); Hussaini et al. (2020)). Isomers typically fall into two main categories: structural isomers, which share the same chemical formula but display distinct atom connectivity, and spatial isomers, which share the same topology graph but diverge in their three-dimensional arrangement (see Appendix D.1). These complex isomers require years of expertises in chemical bonding and spatial relationships to distinguish. Definitely, this is an opportunity to enhance the mentioned understanding of molecular structures, behaviors, and functions, through MLMMA model, incorporating atomic-level alignment, referring to IE-Meta-MMA.

In view of these challenges and opportunities, we propose a novel framework **K-M3AID** (Multil-Level Multimodal Alignment with Knowledge-Guided Instance-Wise Discrimination) incorporating RS-MMA and IE-Meta-MMA, to solve challenging Nuclear Magnetic Resonance (NMR) (Slichter (2013)) spectral alignment task in chemistry (see Figure 1). The overview of our K-M3AID framework is a dual-coordinated contrastive learning architecture, which contains three key components: RS-MMA Module, IE-Meta-MMA Module, and Communication Channel. RS-MMA module establishes the correspondences of molecules with their individual $^{13}$C NMR spectra. Each molecule, with its unique arrangement of atoms and bonding patterns, gives rise to a distinct spectral signature. Thus, we adapt simple cross entropy loss for contrastive learning in RS-MMA module. IE-Meta-MMA module aligns each C atom within the molecules with their signals on the spectrum. In contrast to the diverse and distinctive molecular spectral signatures, many atoms exhibit chemical symmetric and magnetic equivalence within the same molecule, corresponding to the same signals. Meanwhile, atoms with different local surroundings can still presents significant similarity on the spectrum, which introduces heightened level of complexity. In view of these complex scenarios, we come up with knowledge-guided instance wise discrimination based contrastive learning in IE-Meta-MMA module (see Figure 2).

In summary, our contribution comprises three major aspects: *Conceptually:* We integrate IE-Meta-MMA into MLMMA framework, which which facilitates rapid adaptation and enhances the efficiency of learning for multimodal zero-shot tasks. *Methodologically:* We present knowledge-guided instance wise discrimination for cross-modal contrastive learning, which take advantage of continuous and domain-specific features with natural ordering. To the best of our knowledge, this is the first to demonstrate knowledge-guided instance wise discrimination based cross-modal contrastive learning. *Empirically:* We demonstrate the effectiveness of K-M3AID in multiple zero-shot tasks: molecular and atomic alignment, spectrum to molecules retrieval, and isomer recognition.

## 2 RELATED WORK

In general, MLMMA involves three key techniques: multimodal contrastive learning, instance-wise discrimination and meta-alignment.

**Multimodal Contrastive Learning Mechanism:** The paradigm, exemplified by models like CLIP (Contrastive Language-Image Pretraining) (Radford et al. (2021); Li et al. (2021)), accommodates scenarios featuring multiple data modalities. It simultaneously acquires representations for both text and images through two pre-trained unimodal encoders, maps embeddings into a joint space via complemented projection layers, and aligns them through contrastive loss. The overall picture

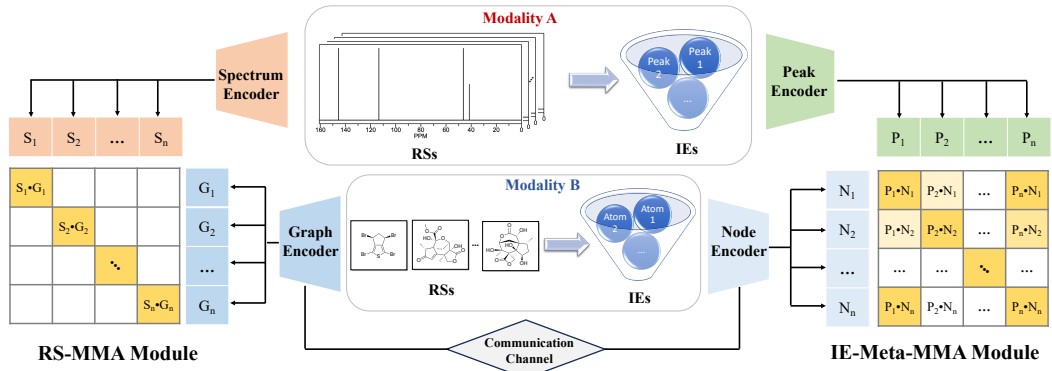

Figure 1: The Architecture of K-M3AID. RSs refers to molecular spectrum and molecules, IEs refers to peaks and atoms. S for spectrum embedding, G for graph embedding, P for peak embedding and N for node embedding.

of CLIP is an end-to-end mechanism, which typically exhibits a symmetric gradient flow in the training process.

**Multimodal Instance-Wise Discrimination:** Instance discrimination (Le-Khac et al. (2020); Zolfaghari et al. (2021); Morgado et al. (2021); Liu et al. (2023)), as a form of self-supervised learning, distinguishes individual instances without explicit class labels. Moving into multimodal contrastive learning, it can be categorized into two general approaches: strong-pair-based (van den Oord et al. (2019); Jaiswal et al. (2021); Liu et al. (2023)) and weak-pair-based (Salakhutdinov & Hinton (2007); Frosst et al. (2019); Liang et al. (2021)) instance-wise discrimination. The strong-pair-based NCE method enforces a precise one-to-one correspondence for real samples with artificially generated noise samples. An example of positive pair can be a noise-added picture of zebra with the text description of zebra. Instead of one-to-one correspondences, weak-pair-based approach relaxes the positive pairs to more boarder semantic correspondences. An example of positive pair can be a picture of zebra with the text description of horse, but not with the text description of tiger.

**Multimodal Meta-Alignment:** As viewed through the lenses of intermediate-level alignment and irreducible element-level alignment, multimodal meta-alignment represents a multifaceted approach to ensuring organizational coherence and effectiveness (Ma et al. (2022)). Exemplary instances of intermediate-level meta-alignment, as seen in works like Cross-Modal Generalization (Chen et al. (2017); Li et al. (2020); Liang et al. (2021); Zhang et al. (2021)) and Livestreaming Product Recognition (Yang et al. (2023)), typically function at both the objective level and the patch level. The exploration of multimodal meta-alignment at the level of irreducible element remains relatively underdeveloped in the current landscape.

## 3 OUR METHOD

In this section, we firstly present the architecture of K-M3AID framework, an end-to-end system designed for MLMMA. Then, we introduce the constrastive learning loss in K-M3AID along with the principles of Knowledge-Guided Instance-Wise Discrimination.

### 3.1 ARCHITECTURE

The K-M3AID framework is a dual-CLIP architecture (see Figure 1), which consists of three critical components: RS-MMA module, IE-Meta-MMA module and comunication channel. The RS-MMA module adapts a gradient-asymmetric CLIP mechanism. While two unimodal encoders work in conjunction, only the from-scratch graph encoder (GIN, Xu et al. (2018)) undergoes dynamic training throughout the process, the pre-trained spectrum encoder (Yang et al. (2021)) remains fixed. Both encoders are complemented by dedicated projection layers, which facilitate the mapping of embeddings into a joint space. The IE-Meta-MMA module adapts a gradient-symmetric CLIP mechanism. It is equipped with two from-scratch unimodal encoders, node encoder and peak encoder, as well as

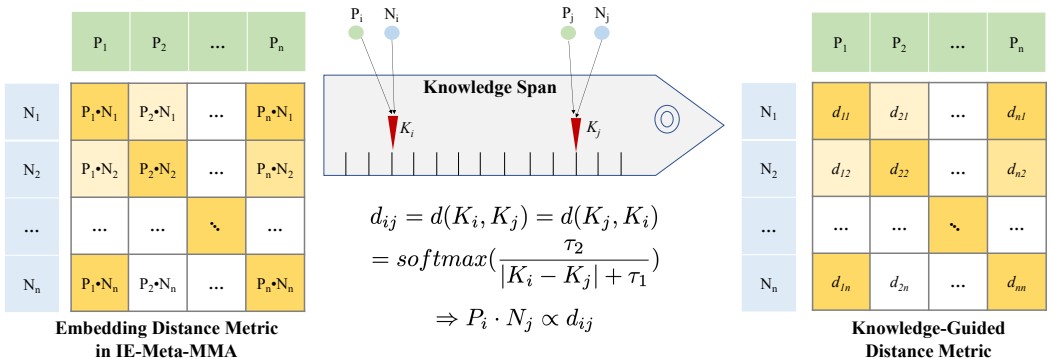

Figure 2: Knowledge Span Mechanism. $K_i$ and $K_j$ represents the corresponding knowledge span label for $i^{th}$ and $j^{th}$ items.

their dedicated projection layers. The graph encoder in RS-MMA module shares part of the weights with node encoder in IE-Meta-MMA module, serving as the communication channel. (See the detail features of respective encoders in Appendix B)

## 3.2 CONTRASTIVE LEARNING LOSS

The synergy between these two modules is pivotal, collectively contributing to our loss function, expressed as

$$L = CL_{RS} + CL_{IE}, \tag{1}$$

where $CL_{RS}$ represents the contrastive learning loss in the RS-MMA module by Equation 3, and $CL_{IE}$ the contrastive learning loss in IE-Meta-MMA module by Equation 6.

Let $i$ denote the $i^{th}$ reducible substance, and $j$ denote the $j^{th}$ reducible substance. Then $x_i$ denotes the raw input in modality A for the $i^{th}$ reducible substance and $y_j$ denotes the raw input in modality B for the $j^{th}$ reducible substance. Suppose $f_x(\cdot)$ represent the encoder function for modality A, and $f_y(\cdot)$ denote the encoder function for modality B. In RS-MMA module, these two unimodal encoder functions, should map $x_i$ and $y_j$ to a proximate location in the joint embedding (inter-modality) if $i = j$.

$$CL_{RS}(i) = \log \frac{e^{\delta(x_i, y_i)}}{\sum_{1 \leq j \leq N} e^{\delta(x_i, y_j)}} \tag{2}$$

$$= \log(\text{softmax}(\delta(x_i, y_i)) \tag{3}$$

Where $\delta(x_i, y_j) = \left( f_x(x_i)^T \cdot f_y(y_j) \right)$, $N$ is the total number of reducible substances from the current batch.

Thus, the total $CL_{RS}$ is expressed as following:

$$CL_{RS} = \frac{1}{N} \sum_{1 \leq i \leq N} CL_{RS}(i) \tag{4}$$

This design for the loss aims to match the same reducible substance cross different modalities.

## 3.3 KNOWLEDGE-GUIDED INSTANCE WISE DISCRIMINATION CONTRASTIVE LEARNING

**Knowledge Span**, in which we define as some continuous and domain-specific feature exhibiting natural ordering and offering guidance, can potentially offer valuable insights into the contrastive learning process. As such, we propose a novel and general approach to contrastive learning called

knowledge-guided instance-wise discrimination. This approach expands the scope of contrastive learning from confined comparisons (pre-determined negative and positive pairs) to unrestricted comparisons (no need for pre-determination). This extension removes the necessity of explicitly defining such pairs, thus mitigating the potential introduction of human bias.

Suppose $\mathcal{M}$ is the set of irreducible elements in the reducible substances. $\mathcal{A} \subset \mathbb{R}^{d_1}$ is the set of tunable irreducible elements' embeddings in modality A, $\mathcal{B} \subset \mathbb{R}^{d_1}$ is the set of tuable irreducible elements' embeddings in modality B, and $\mathcal{K} \subset \mathbb{R}^{d_2}$ is the corresponding fixed **knowledge span label** that can guide the relative distance learning between components in $\mathcal{A}$ and $\mathcal{B}$. Thus, the size of $\mathcal{A}, \mathcal{B}, \mathcal{K}$ are $|\mathcal{M}|$, respectively.

Let $\mathcal{A}_i$ be the $i^{th}$ irreducible element embedding of $\mathcal{A}$, and $\mathcal{B}_j$ be the $j^{th}$ irreducible element embedding of $\mathcal{B}$. We define the distance function between $\mathcal{A}_i$ and $\mathcal{B}_j$ as $d_E(\mathcal{A}_i, \mathcal{B}_j) = \mathcal{A}_i \cdot \mathcal{B}_j \to \mathbb{R}^+$, and calibration function $d(\mathcal{K}_i, \mathcal{K}_j) \to \mathbb{R}^+$ with a monotonic property and constraint $\sum_{j=1}^{|\mathcal{M}|} d(\mathcal{K}_i, \mathcal{K}_j) = 1$, in which $\mathcal{K}_i$ and $\mathcal{K}_j$ serve as the designated Knowledge Span Label. We introduce the **Knowledge Span Guided Loss (KSGL)** as follows:

$$KSGL(i) = - \sum_{1 \leq j \leq |\mathcal{M}|} d(\mathcal{K}_i, \mathcal{K}_j) \log \frac{e^{d_E(\mathcal{A}_i, \mathcal{B}_j)}}{\sum_{1 \leq k \leq |\mathcal{M}|} e^{d_E(\mathcal{A}_i, \mathcal{B}_k)}} \tag{5}$$

$$= - \sum_{1 \leq j \leq |\mathcal{M}|} d(\mathcal{K}_i, \mathcal{K}_j) \log(\text{softmax}(d_E(\mathcal{A}_i, \mathcal{B}_j))) \tag{6}$$

In particular, when it reaches ideal optimum, $d(\mathcal{K}_i, \mathcal{K}_j)$ and $d_E(\mathcal{A}_i, \mathcal{B}_j)$ reaches the following relation:

$$d(\mathcal{K}_i, \mathcal{K}_j) = \text{softmax}(d_E(\mathcal{A}_i, \mathcal{B}_j)) \tag{7}$$

For detail proof, please refer to Appendix A As a result, the corresponding $CL_{IE}$ is expressed as following:

$$CL_{IE} = \frac{1}{|\mathcal{M}|} \sum_{1 \leq i \leq |\mathcal{M}|} KSGL(i) \tag{8}$$

### 3.4 CHOSEN KNOWLEDGE SPAN-PPM

$^{13}$C NMR uncovers molecular structures by providing the chemical environments of carbon atoms and their magnetic responses to external fields, and quantifies these features in parts per million (ppm) relative to a reference compound like tetramethylsilane (TMS), simplifying comparisons across experiments. Thus, continuous peak positions, measured in ppm, can serve as a robust knowledge span to facilitate instance-wise discrimination for this contrastive learning task.

For IE-Meta-MMA module in the case of ppm guide, $\mathcal{A}$ is the set of learned node embeddings for Carbon atoms and $\mathcal{B}$ is the set of learned peak embeddings for respective Carbon atoms, $\mathcal{K}$ is the set of ppm value for each corresponding Carbon atom in $\mathcal{A}$ and $\mathcal{B}$. Suppose $ppm_i$ is the ppm for the $i^{th}$ Carbon Atom and $ppm_j$ is the corresponding ppm for $j^{th}$ peak. $d(\cdot, \cdot)$ is then defined as following:

$$d(\mathcal{K}_i, \mathcal{K}_j) = d(ppm_i, ppm_j) = softmax(\frac{\tau_2}{|ppm_i - ppm_j| + \tau_1}) \tag{9}$$

where $\tau_1$ and $\tau_2$ are temperature hyper-parameter. For further discussion of selection about $\tau_1$ and $\tau_2$, please refer to Appendix C.2. Then, the final form of contrastive loss for irreducible atom level according to Equation 6 is as following:

$$KSGL(i) = -\frac{1}{|\mathcal{M}|} \sum_{1 \leq j \leq |\mathcal{M}|} d(ppm_i, ppm_j) \log \frac{e^{d_E(\mathcal{A}_i, \mathcal{B}_j)}}{\sum_{1 \leq k \leq |\mathcal{M}|} e^{d_E(\mathcal{A}_i, \mathcal{B}_k)}} \tag{10}$$

# 4 EXPERIMENTS

## 4.1 EXPERIMENTAL SETUP

### 4.1.1 DATASETS AND TASKS

In the training of K-M3AID model, the dataset comprises over 20,000 data points sourced from nmrshiftdb2 (Steinbeck et al. (2003)). In this dataset, molecule are aligned with their respective $^{13}$C NMR spectra, and atomic alignments with peaks are also included. The quality of the dataset was further validated by experienced organic chemists. In zero-shot isomer recognition task, the dataset were never appeared in the training dataset, and each of the isomer groups contains at least 10 molecules, which are structural isomers or spatial isomers to each other (see details for isomers in Appendix D). In the task of zero-shot molecular retrieval, 1000 spectra (never appeared in training dataset) was used, the molecules were collected over 1 million from Pub-Chem (Kim et al. (2023)), and randomly chosen for the experiments.

### 4.1.2 BALANCE OF $CL_{RS}$ AND $CL_{IE}$

In order to gain insights into how the interplay between $CL_{RS}$ and $CL_{IE}$ impacts on both molecular-level accuary in RS-MMA and atomic-level alignment accuracy in IE-Meta-MMA, we introduced a parameter $\alpha$ to adjust the weights of $CL_{RS}$ and $CL_{IE}$:

$$L = \alpha * CL_{RS} + (1 - \alpha) * CL_{IE}, \tag{11}$$

and conducted a series of studies regarding $\alpha$, where $0 \leq \alpha \leq 1$ (see Table 1 and Appendix C.3).

To begin, when utilizing $CL_{RS}$ at full capacity with $\alpha = 1$, the accuracy of molecular alignment reaches approximately 94.6%. However, the accuracy of atomic alignment is approximately 17.6%, as the $CL_{IE}$ for atomic alignment was omitted. Conversely, when neglecting $CL_{RS}$ with $\alpha = 0$ and relying solely on $CL_{IE}$, the accuracy for molecular alignment experiences a dramatic decrease to merely around 0.7%, but the accuracy for atomic alignment significantly improves to approximately 90.4%. These findings imply that the success of molecular alignment can offer certain degree of guidance for atomic alignment, but the fulfillment of on atom alignment proves inadequate for directing molecular alignment in the desired direction. Thus, we continue to alter the value of $\alpha$ from 0 to 1, the accuracy for molecule alignment undergoes an initial increase followed by a subsequent decline, finding its optimal performance of 95.5% at $\alpha = 0.5$. On the other hand, the accuracy of atomic alignment remains stable at approximately 90% for $\alpha$ ranging from 0 to 0.2. However, when $\alpha$ exceeds 0.2, a decrease can be observed on the accuracy of atomic alignment. It indicates an excessive emphasis on molecular alignment leads to a decrease on the performance of atomic alignment. Thus, we decide $\alpha = 0.2$ is the optimal setting for the following experiments.

Table 1: Balancing $CL_{RS}$ and $CL_{IE}$ via $\alpha$ ablation study to evaluate accuracy with $epochs = 200$, $\tau_1 = 10^1$ and $\tau_2 = 10^{-5}$.

| $\alpha$ | 0.00 | 0.10 | 0.20 | 0.50 | 0.80 | 0.90 | 1.00 |
|---|---|---|---|---|---|---|---|
| RS-MMA | 0.7±0.3 | 94.3±0.1 | 94.7±0.4 | **95.5±0.4** | 95.1±0.2 | 94.8±0.5 | 94.6±0.4 |
| IE-Meta-MMA | 90.4±0.2 | 90.3±0.3 | **90.3±0.1** | 89.6±0.0 | 86.3±0.8 | 83.7±0.4 | 17.6±3.1 |

## 4.2 RESULTS OF K-M3AID

### 4.2.1 PERFORMANCE ON RS-MMA

K-M3AID model achieves an validation accuracy of above 94% for molecular-level alignments in RS-MMA module after 200 epochs. Subsequently, we evaluate the capability of our trained model by conducting retrieval of specific molecule based on the given spectrum (**Spec2Mol**) on various dataset sizes (see Table 2). For a molecular dataset containing 100 entries, the K-M3AID model consistently achieves approximately 95% accuracy in retrieval within the top 1%, 5%, 10%, and 25% of results. For a molecular dataset with a size of $10^3$, the K-M3AID model consistently achieves retrieval accuracies more than 97.0% within the top 5%, 10%, and 25%. In the case of a molecular dataset with a size of $10^4$, the K-M3AID model maintains an accuracy level of 85.9% for the top

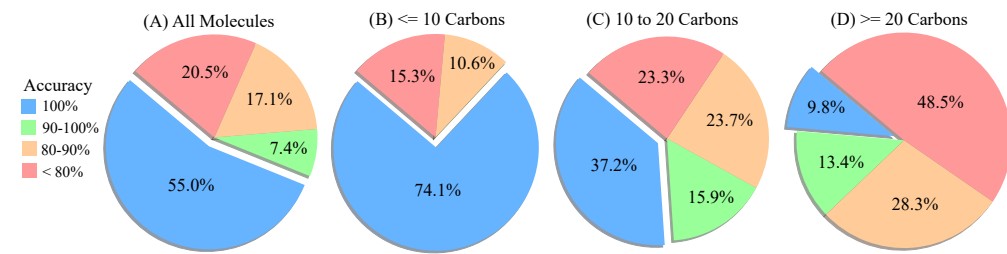

Figure 3: Atomic Alignment in IE-Meta-MMA Module

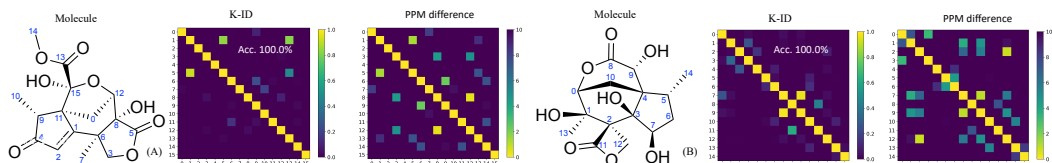

Figure 4: Examples of Zero-shot Atomic Alignment for Complex Natural Products. Yellow cells in the PPM differerence represent the ground truth alignment. For exatra cases, please refer to Appendix E

10% retrieval and approximately 93% for the top 25% retrieval. Notably, for a molecular dataset with a size of up to $10^5$ entries, the K-M3AID model attains an average accuracy as 53.1% for the top 10% retrieval and 68.2% for the top 25% retrieval. These results set the K-M3AID model apart from other methods, making it an exceptional choice in such scenarios.

Table 2: Zero-shot Spec2Mol task on molecular datasets with different number of molecules

| Accuracy | $10^2$ | $10^3$ | $10^4$ | $10^5$ | $10^6$ |
|---|---|---|---|---|---|
| Top 1(%) | 95.4±0.6 | 77.3±2.5 | 44.7±3.3 | 16.2±3.6 | 4.4±2.3 |
| Top 5(%) | 100.0±0.2 | 97.3±0.4 | 77.5±2.7 | 40.2±4.8 | 12.3±4.8 |
| Top 10(%) | 100.0±0.1 | 99.1±0.5 | 85.9±1.9 | 53.1±4.2 | 18.5±5.3 |
| Top 25(%) | 100.0±0.1 | 99.8±0.3 | 93.3±0.8 | 68.2±3.1 | 29.7±6.4 |

### 4.2.2 PERFORMANCE ON IE-META-MMA

K-M3AID model achieves an validation accuracy of above 90% for atomic-level alignment in IE-Meta-MMA module after 200 epochs as shown by Figure 3. Within the validation set from 5-fold experiments, there are 12771 molecules containing fewer than 10 carbon atoms, 7043 molecules with carbon atom counts between 10 and 20, and 1138 molecules with more than 20 carbon atoms. Specifically, our model achieves 100% accuracy in 74.1% of the molecules containing fewer than 10 C atoms. For molecules with 10 to 20 C atoms, our model achieves 100% accuracy in 37.2% of cases. Furthermore, it attains an accuracy exceeding 80% in more than 50% of the molecules containing more than 20 C atoms.

In complex natural product molecules, it is a common situation that the local contents of some atoms within the same molecule exhibit a high degree of similarity. It gives rise to challenges for the atomic alignment, as some atoms correspond to ppm values in close proximity. However, our K-M3AID model is capable of recognizing each of the atoms with effective learnt embeddings and deciphering the correspondences among the atoms and the peaks at zero-shot. Two complex natural product molecules with multiple rings (4 and 4, respectively) and multiple chiral centers (6 and 8, respectively) are taken to showcase the effectiveness of atomic alignment (see Figure 4).

### 4.3 COMPARISON TO EXISTING INSTANCE WISE DISCRIMINATION APPROACHES

In K-M3AID, knowledge-guided instance-wise discrimination (K-ID) is adapted into IE-Meta-MMA module. As strong-pair-based (SP-ID) and weak-pair-based (WP-ID) instance-wise discrim-

ination are general apparoches in contrastive leraning, we replace K-ID with SP-ID and WP-ID to conduct a comparative analysis for the impact of K-ID, SP-ID and WP-ID on the molecular and atomic alignment. SP-ID confines the irreducible elements (atoms and peaks) have a precise match across different modalities with the sole correct pairs established in the training process. However, WP-ID extends the scope from one precise match to multiple matches within the chosen threshold set for the distance of their corresponding ppm. In this context, the mathematical definition of strong pair and weak pair are given as follows:

$$\text{Strong Pair: } |ppm_i - ppm_j| = 0, \tag{12}$$

$$\text{Weak Pair: } |ppm_i - ppm_j| \leq th, \tag{13}$$

where $A_i \in A$, which stands for atoms; $P_j \in P$, which stands for peaks; $th$ is the abbreviation of threshold. In addition, $ppm_i$ is the ppm for the $i^{th}$ Carbon Atom and $ppm_j$ is the corresponding ppm for $j^{th}$ peak.

Table 3: Validation accuracy of SP-ID-based and WP-ID-based models with $epochs = 200$ and $\alpha = 0.2$.

| Method | SP-ID | WP-ID(th=1) | WP-ID(th=2) | WP-ID(th=5) | WP-ID(th=10) | K-ID |
|---|---|---|---|---|---|---|
| RS MMA | 93.5±0.6 | 91.3±0.8 | 90.6±0.4 | 90.3±0.6 | 88.4±1.4 | **94.7±0.4** |
| IE MMA | 89.3±0.4 | 83.7±0.6 | 83.2±0.2 | 79.8±0.5 | 66.1±2.5 | **90.2±0.1** |

### 4.3.1 COMPARISON ON RS-MMA

K-ID outperforms SP-ID and WP-ID in molecular-level alignment in RS-MMA module (see Table 3). K-ID enables the molecular-level alignment to achieve an validation accuracy rate around 94%, 1 to 6% higher than other approaches. Meanwhile, K-ID distinguishes itself prominently over SP-ID and WP-ID approaches in the task of zero-shot isomer recognition by giving 100% accuracy for multiple groups of isomers (see Table 4). Furthermore, as for zero-shot Spec2Mol task, along the size of the molecular dataset increases, our K-ID-based model consistently exhibits superiority over existing methods such as SP-ID and WP-ID (see Table 2 and Table 5). These empirical findings underscore the benefits of K-ID based IE-Meta-MMA in the context of Spec2Mol, indicating its positive impact to RS-MMA.

Table 4: Zero-Shot Isomer Recognition Accuracy with SP-ID-based, WP-ID-based and K-ID-based models. For detail demo of $C_7H_{11}NO_3$, please refer to Appendix D.2

| Formula | #Isomers | SP-ID (%) | WP-ID (th=1) (%) | K-ID(%) |
|---|---|---|---|---|
| $C_4H_6O$ | 15 | 86.7 | 86.7 | **100.0** |
| $C_9H_9N$ | 15 | 86.7 | 80.0 | **100.0** |
| $C_7H_{11}NO_3$ | 14 | 78.6 | 85.7 | **100.0** |
| $C_6H_{13}NO$ | 23 | 91.3 | 91.3 | **100.0** |
| $C_8H_7NO_4$ | 13 | 92.3 | 84.6 | **100.0** |
| $C_{15}H_{24}O$ | 16 | 93.8 | 93.8 | **100.0** |
| $C_{11}H_{14}$ | 10 | 90.0 | 80.0 | **100.0** |
| $C_7H_{15}NO$ | 14 | 85.7 | 85.7 | **100.0** |
| $C_{10}H_{16}O_2$ | 26 | 92.3 | 84.6 | **100.0** |
| $C_8H_{15}N$ | 11 | 81.8 | 90.9 | **100.0** |

### 4.3.2 COMPARISON ON IE-META-MMA

K-ID pushes the validation accuracy of atomic-level alignment above 90%, 1 to 24% higher than SP-ID and WP-ID approaches in IE-Meta-MMA module (see Table 2 and Table 3). This superiority arises from the inherent limitations of both strong and weak pair definitions, which is failing to precisely calibrate the diverse relationships among the elements. This finding is further supported by the significant decreases in the accuracy of atomic alignment as the threshold of weak pair increases. The limitation of either SP-ID or WP-ID becomes notably significant in the following two scenarios: 1) when local contents of some atoms exhibit a high degree of similarity; 2) when some atoms exhibit symmetric mapping within the same molecule.

Table 5: Zero-Shot Spec2Mol Accuracy with SP-ID and WP-ID on Pub-Chem Database

| Method | Accuracy | $10^2$ | $10^3$ | $10^4$ | $10^5$ | $10^6$ |
|---|---|---|---|---|---|---|
| SP-ID | Top 1(%) | 95.3±0.8 | 78.6±2.7 | 35.8±3.8 | 12.9±1.6 | 3.4±0.9 |
| | Top 5(%) | 95.4±0.1 | 77.3±0.7 | 44.7±2.3 | 16.2±2.4 | 4.4±1.5 |
| | Top 10(%) | 100.0±0.0 | 97.3±0.7 | 77.5±2.3 | 40.2±2.4 | 12.3±1.5 |
| | Top 25(%) | 100.0±0.0 | 99.1±0.2 | 85.9±1.0 | 53.1±3.0 | 18.5±1.8 |
| WP-ID(th=1) | Top 1(%) | 92.9±0.6 | 71.7±1.0 | 32.7±1.3 | 10.7±0.5 | 3.6±0.7 |
| | Top 5(%) | 99.6±0.1 | 93.8±0.8 | 63.9±1.5 | 29.3±1.5 | 10.2±1.2 |
| | Top 10(%) | 99.9±0.0 | 97.1±0.4 | 76.8±0.7 | 39.3±0.9 | 15.7±1.5 |
| | Top 25(%) | 100.0±0.0 | 99.1±0.2 | 88.2±0.6 | 55.7±1.1 | 26.5±2.0 |

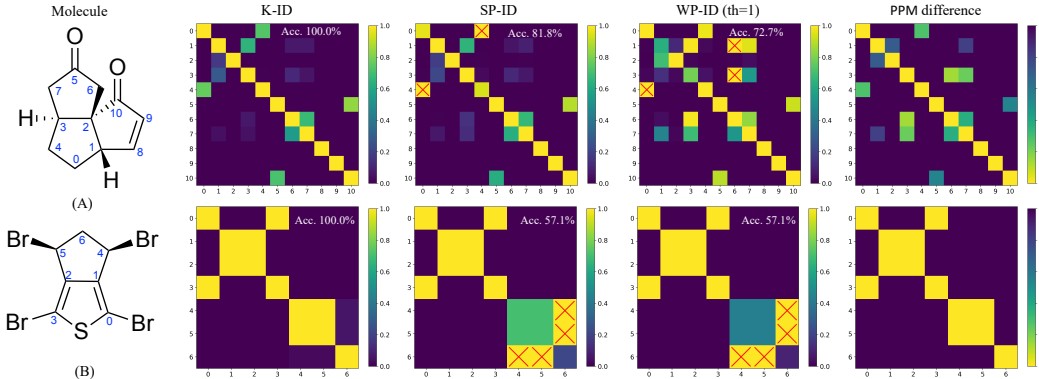

Figure 5: Case study of IE-Meta-MMA. Yellow cells in the PPM differerence represent the ground truth alignment, and red cross represents the wrong alignment.

In the former scenario, exemplified by molecular A in Figure 5, atom 0 and atom 4 are secondary carbons (attaching to 2 carbons and 2 hydrogens), nearly symmetric on the same 5-member ring, corresponding to the ppm of 27.0 and 29.8, respectively. The similar local content of these two atoms fools SP-ID and WP-ID. Meanwhile, atom 1 and atom 3 are tertiary carbons (attaching to 3 carbons and 1 hydrogen), nearly symmetric on the same 5-member ring, corresponding to the ppm of 54.5 and 44.1, respectively. Only WP-ID fails to distinguish and align them. In the later scenario, exemplified by molecular B in Figure 5, there exist instances one-to-one and one-to-many for atomic-level alignment within the molecular configuration. Both SP-ID and WP-ID method misaligns certain atoms with other atoms with small ppm differences (less than 3 ppm in this case), rather than aligning them with themselves or their symmetric counterparts. In contrast, K-ID approach excels in both scenario by discerning each one of the atoms, which is attributed to the full utilization of ppm difference distance learning (see additional examples in Appendix E).

## 5 CONCLUSION AND FUTURE WORK

In this paper, we introduced the Knowledge-Guided Multi-Level Multimodal Alignment with Instance-Wise Discrimination (K-M3AID) framework, incorporating RS-MMA and IE-Meta-MMA. Its effectiveness was demonstrated through multiple zero-shot tasks: molecular and atomic alignment, Spec2Mol and isomer recognition. And we highlighted the significance of knowledge-guided instance-wise discrimination via a few metrics and case studies. Furthermore, we presented experiments aimed at accommodating the dynamic interactions between RS-MMA and IE-Meta-MMA. While our framework achieves an atomic-level alignment overall accuracy of 100% for 55% of cases, it drops significantly to 9.8% when dealing with molecules containing more than 20 carbon atoms. Currently, our graph encoder is implemented on 2D-molecular graph with basic node and edge features, potentially limiting its ability to produce precise node embeddings to distinguish atoms in the extremely complex scenarios. In the future developments, the incorporation of a 3D-based graph holds a great potential to improve performance in this regard.

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

# Appendix

## A   REVISITING KNOWLEDGE SPAN GUIDED LOSS

**Theorem 1** (Knowledge Span Guided Loss). *Suppose $\mathcal{M}$ is the set of irreducible elements in the reducible substances. $\mathcal{A} \subset \mathbb{R}^{d_1}$ is the set of tunable irreducible elements' embeddings in modality A, $\mathcal{B} \subset \mathbb{R}^{d_1}$ is the set of tuable irreducible elements' embeddings in modality B, and $\mathcal{K} \subset \mathbb{R}^{d_2}$ is the corresponding fixed **knowledge span label** that can guide the relative distance learning between components in $\mathcal{A}$ and $\mathcal{B}$. Thus, the size of $\mathcal{A}$, $\mathcal{B}$, $\mathcal{K}$ are $|\mathcal{M}|$, respectively.*

*Let $\mathcal{A}_i$ be the $i^{th}$ irreducible element embedding of $\mathcal{A}$, and $\mathcal{B}_j$ be the $j^{th}$ irreducible element embedding of $\mathcal{B}$. We define the distance function between $\mathcal{A}_i$ and $\mathcal{B}_j$ as $d_E(\mathcal{A}_i, \mathcal{B}_j) = \mathcal{A}_i \cdot \mathcal{B}_j \to \mathbb{R}^+$, and calibration function $d(\mathcal{K}_i, \mathcal{K}_j) \to \mathbb{R}^+$ with a monotonic property and constraint $\sum_{j=1}^{|\mathcal{M}|} d(\mathcal{K}_i, \mathcal{K}_j) = 1$, in which $\mathcal{K}_i$ and $\mathcal{K}_j$ serve as the designated Knowledge Span Label. We introduce the **Knowledge Span Guided Loss (KSGL)** as follows:*

$$KSGL(i) = - \sum_{1 \le j \le |\mathcal{M}|} d(\mathcal{K}_i, \mathcal{K}_j) \log \frac{e^{d_E(\mathcal{A}_i, \mathcal{B}_j)}}{\sum_{1 \le k \le |\mathcal{M}|} e^{d_E(\mathcal{A}_i, \mathcal{B}_k)}} \quad (A.1)$$

$$= - \sum_{1 \le j \le |\mathcal{M}|} d(\mathcal{K}_i, \mathcal{K}_j) \log(softmax(d_E(\mathcal{A}_i, \mathcal{B}_j))) \quad (A.2)$$

*Proof.* In order to optimize the loss $KSGL(i)$, we need to set the following partial derivative to be 0 for each $d_E(\mathcal{A}_i, \mathcal{B}_j)$ with $1 \le j \le |\mathcal{M}|$. Here are the detail process:

$$\frac{\partial KSGL(i)}{\partial d_E(\mathcal{A}_i, \mathcal{B}_j)} = \underbrace{\frac{\partial}{\partial d_E(\mathcal{A}_i, \mathcal{B}_j)} \left( -d(\mathcal{K}_i, \mathcal{K}_j) \log \frac{e^{d_E(\mathcal{A}_i, \mathcal{B}_j)}}{e^{d_E(\mathcal{A}_i, \mathcal{B}_j)} + \sum_{k \ne j} e^{d_E(\mathcal{A}_i, \mathcal{B}_k)}} \right)}_{\text{When the numerator includes } e^{d_E(\mathcal{A}_i, \mathcal{B}_j)}}$$

$$+ \underbrace{\frac{\partial}{\partial d_E(\mathcal{A}_i, \mathcal{B}_j)} \left( \sum_{k \ne j} -d(\mathcal{K}_i, \mathcal{K}_k) \log \frac{e^{d_E(\mathcal{A}_i, \mathcal{B}_k)}}{e^{d_E(\mathcal{A}_i, \mathcal{B}_j)} + \sum_{k \ne j} e^{d_E(\mathcal{A}_i, \mathcal{B}_k)}} \right)}_{\text{When the numerator does not include } e^{d_E(\mathcal{A}_i, \mathcal{B}_j)}}$$

$$= -(d(\mathcal{K}_i, \mathcal{K}_j) - d(\mathcal{K}_i, \mathcal{K}_j) \cdot softmax(d_E(\mathcal{A}_i, \mathcal{B}_j))$$

$$- \sum_{k \ne j} d(\mathcal{K}_i, \mathcal{K}_k) \cdot softmax(d_E(\mathcal{A}_i, \mathcal{B}_j))$$

$$= - \left( d(\mathcal{K}_i, \mathcal{K}_j) - (d(\mathcal{K}_i, \mathcal{K}_j) + \sum_{k \ne j} d(\mathcal{K}_i, \mathcal{K}_k)) \cdot softmax(d_E(\mathcal{A}_i, \mathcal{B}_j)) \right)$$

Since $\sum_{l=1}^{|\mathcal{M}|} d(\mathcal{K}_i, \mathcal{K}_l) = 1$, we can further simplify it as

$$\frac{\partial KSGL(i)}{\partial d_E(\mathcal{A}_i, \mathcal{B}_j)} = -(d(\mathcal{K}_i, \mathcal{K}_j) - softmax(d_E(\mathcal{A}_i, \mathcal{B}_j))$$

In order to optimize, we need to set the respective partial derivative to be 0:

$$\frac{\partial KSGL(i)}{\partial d_E(\mathcal{A}_i, \mathcal{B}_j)} = -(d(\mathcal{K}_i, \mathcal{K}_j) - softmax(d_E(\mathcal{A}_i, \mathcal{B}_j)) = 0$$

In addition, the corresponding second partial derivative denoted as $\frac{\partial KSGL(i)}{\partial d_E^2(\mathcal{A}_i, \mathcal{B}_j)}$ manifests as follows:

$$\frac{\partial KSGL(i)}{\partial d_E^2(\mathcal{A}_i, \mathcal{B}_j)} = softmax(d_E(\mathcal{A}_i, \mathcal{B}_j))(1 - softmax(d_E(\mathcal{A}_i, \mathcal{B}_j)))$$

As $\text{softmax}(d_E(\mathcal{A}_i, \mathcal{B}_j))$ takes values within the open interval (0,1), it follows that $\frac{\partial KSGL(i)}{\partial d_E^2(\mathcal{A}_i, \mathcal{B}_j)}$ is always positive. Consequently, the pinnacle of optimization emerges as a global minimum. Furthermore, when it comes to optimum:

$$d(\mathcal{K}_i, \mathcal{K}_j) = \text{softmax}(d_E(\mathcal{A}_i, \mathcal{B}_j))$$

$$d_E(\mathcal{A}_i, \mathcal{B}_j) = \log(d(\mathcal{K}_i, \mathcal{K}_j)) + \log\left(\sum_{1 \leq l \leq |\mathcal{M}|} e^{d_E(\mathcal{A}_i, \mathcal{B}_l)}\right)$$

It is easy to show that when it reaches optimum, $d_E(A_i, B_j)$ is consistent with Knowledge Span Guidance $d(\mathcal{K}_i, \mathcal{K}_j)$. Without loss of generosity, suppose $d(\mathcal{K}_i, \mathcal{K}_j) > d(\mathcal{K}_i, \mathcal{K}_{j'})$ :

$$
\begin{aligned}
d_E(\mathcal{A}_i, \mathcal{B}_j) - d_E(\mathcal{A}_i, \mathcal{B}_{j'}) &= \log(d(\mathcal{K}_i, \mathcal{K}_j)) + \log\left(\sum_{1 \leq l \leq |\mathcal{M}|} e^{d_E(\mathcal{A}_i, \mathcal{B}_l)}\right) \\
&\quad - \left(\log(d(\mathcal{K}_i, \mathcal{K}_{j'})) + \log\left(\sum_{1 \leq l \leq |\mathcal{M}|} e^{d_E(\mathcal{A}_i, \mathcal{B}_l)}\right)\right) \\
&= \log(d(\mathcal{K}_i, \mathcal{K}_j)) - \log(d(\mathcal{K}_i, \mathcal{K}_{j'})) \\
&= \log\left(\frac{d(\mathcal{K}_i, \mathcal{K}_j)}{d(\mathcal{K}_i, \mathcal{K}_{j'})}\right) > 0
\end{aligned}
$$

$\square$

## B  ENCODER

The following Table B.1 explains the features used in respective encoders.

Table B.1: Features code for respective encoders in K-M3AID

| Encoder | Feature | Data type |
|---|---|---|
| Graph/Node | Atomic Number (node feature) | Categorical |
| | Chiral tag (node feature) | Categorical |
| | Hybridization (node feature) | Categorical |
| | Bond type (edge feature) | Categorical |
| | Bond direction (edge feature) | Categorical |
| Spectrum | peak intensity | Continuous value |
| | peak position | Continuous value |
| Peak | peak multiplicity | Categorical |
| | peak position | Continuous value |

## C  FURTHER ABLATION STUDY ABOUT PARAMETER CHOICES

### C.1  ABLATION STUDY ABOUT THE CHOICE OF GIN STRUCTURE AND PROJECTION.

We choose GIN(Xu et al. (2018)) as our graph encoder. By Table C.1, "GIN Depth" signifies the number of layers in the GIN, "GIN Embedding Dim" denotes the dimensionality of the embeddings generated by the GIN model, and "Projection Dim" indicates the resulting dimensionality after transforming the GIN-produced embeddings. In particular, the best performance is observed when the GIN model has 5 layers, GIN Embedding Dim is 128, and projection Dim is 512.

### C.2  ABLATION STUDY ABOUT THE CHOICE OF $\tau_1$ AND $\tau_2$.

We also conducted a further ablation study exploring different combinations of $\tau_1$ and $\tau_2$ as shown in Table C.2. For this analysis, we fixed the GIN depth at 5, set the GIN embedding dimensionality

Table C.1: GIN structure and projection ablation study

| GIN Depth | GIN Embedding Dim | Projection Dim | Validation accuracy (%) |
|---|---|---|---|
| 3 | 128 | 128 | 86.6 |
| 3 | 256 | 128 | 86.8 |
| 3 | 512 | 128 | 86.3 |
| 5 | 128 | 128 | 89.4 |
| 5 | 256 | 128 | 89.6 |
| 5 | 512 | 128 | 89.3 |
| 3 | 128 | 256 | 86.6 |
| 3 | 256 | 256 | 86.8 |
| 3 | 512 | 256 | 86.3 |
| 5 | 128 | 256 | 89.4 |
| 5 | 256 | 256 | 89.6 |
| 5 | 512 | 256 | 89.3 |
| 3 | 128 | 512 | 86.6 |
| 3 | 256 | 512 | 86.5 |
| 3 | 512 | 512 | 86.2 |
| 5 | 32 | 512 | 84.0 |
| 5 | 64 | 512 | 87.5 |
| 5 | 128 | 512 | **90.0** |
| 5 | 256 | 512 | 89.4 |
| 5 | 512 | 512 | 88.9 |

to 128, and maintained a projection dimension of 512. Additionaly, the ratio of $CL_{RS}$ referred by Equation 4 to $CL_{IE}$ referred by Equation 8 is 1:1. We observe that the best performance is achieved when $\tau_1 = 10^{-5}$ and $\tau_2 = 10^1$.

Table C.2: Ablation Study about $tau_1$ and $tau_2$. We have 5 layers and 128 dimension as the final representation. In the loss function, $CL_M : CL_A$ = 1:1 after 200 epochs

| $\tau_1$ | $\tau_2$ | Molecular Alignment Accuracy (%) | Atom Alignment Accuracy (%) |
|---|---|---|---|
| $10^{-1}$ | $10^1$ | 94.9 | 89.6 |
| $10^{-1}$ | $10^2$ | 95.2 | 89.8 |
| $10^{-1}$ | $10^3$ | 95.6 | 89.6 |
| $10^{-1}$ | $10^4$ | 95.1 | 88.9 |
| $10^{-1}$ | $10^5$ | 95.0 | 89.3 |
| $10^{-2}$ | $10^1$ | 95.5 | 89.8 |
| $10^{-2}$ | $10^2$ | 94.8 | 89.8 |
| $10^{-2}$ | $10^3$ | 95.4 | 88.8 |
| $10^{-2}$ | $10^4$ | 94.8 | 87.2 |
| $10^{-2}$ | $10^5$ | 95.1 | 89.4 |
| $10^{-3}$ | $10^1$ | 95.0 | 89.2 |
| $10^{-3}$ | $10^2$ | 95.1 | 89.1 |
| $10^{-3}$ | $10^3$ | 95.2 | 89.0 |
| $10^{-3}$ | $10^4$ | 95.3 | 89.7 |
| $10^{-3}$ | $10^5$ | 95.0 | 89.4 |
| $10^{-4}$ | $10^1$ | 95.0 | 89.8 |
| $10^{-4}$ | $10^2$ | 95.1 | 89.7 |
| $10^{-4}$ | $10^3$ | 95.0 | 89.8 |
| $10^{-4}$ | $10^4$ | 95.3 | 89.5 |
| $10^{-4}$ | $10^5$ | 95.1 | 88.4 |
| $10^{-5}$ | $10^1$ | **95.4** | **89.9** |
| $10^{-5}$ | $10^2$ | 95.0 | 89.5 |
| $10^{-5}$ | $10^3$ | 95.8 | 89.6 |
| $10^{-5}$ | $10^4$ | 95.2 | 89.7 |
| $10^{-5}$ | $10^5$ | 95.0 | 89.7 |

### C.3 FURTHER ABLATION STUDY OF ALPHA

As mentioned previously, in order to gain insights into how the interplay between $CL_{RS}$ and $CL_{IE}$ impacts on both molecular-level accuary in RS-MMA and atomic-level alignment accuracy in IE-Meta-MMA, we introduced a parameter $\alpha$ to adjust the weights of $CL_{RS}$ and $CL_{IE}$ by Equation 11. The Figure C.1 about alpha shows more configuration about alpha.

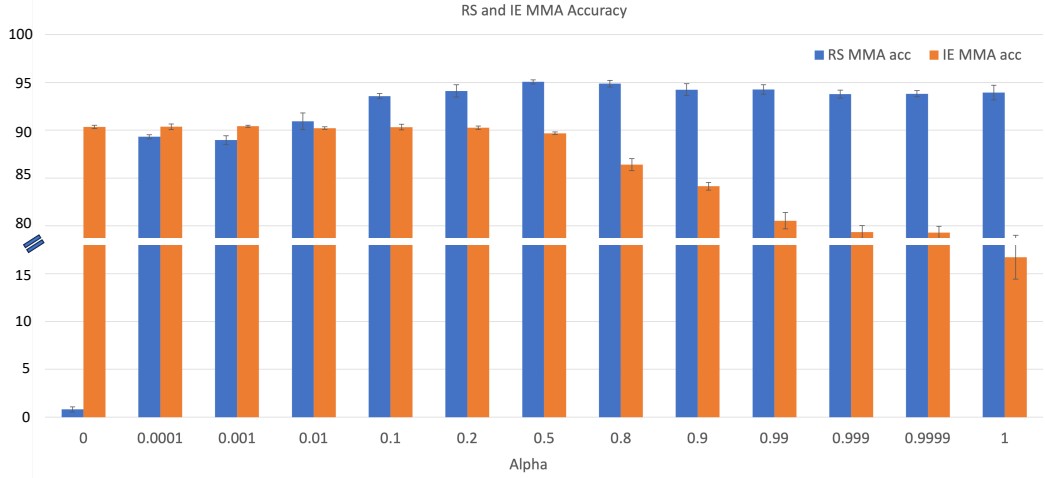

Figure C.1: alpha change

## D MORE DISCUSSION ABOUT ISOMERS

### D.1 BRIEF INTRODUCTION ABOUT ISOMER CATEGORY

Isomers typically fall into two main categories: constitutional (structural) isomers, which share the same chemical formula but display distinct atom connectivity, and stereoisomers (spatial isomers), which share the same topology graph but diverge in their three-dimensional arrangement (see Figure D.1). Constitutional isomers are NMR-variant, meaning that different isomers produce distinct NMR spectrum. In the sub-categories of stereoisomers, enantiomers are NMR-variant, but diastereomers and cis-trans isomers are NMR-variant.

### D.2 ISOMERS GROUP FOR $C_7H_{11}NO_3$

Here is an example for isomer groups. In this isomer group of $C_7H_{11}NO_3$, they all share the same chemical formula in Figure D.2. The first 10 are constitutional (structural) isomers of each other (cycled green), the last 4 are two pairs of diastereomers (cycled brown). Each of these isomers corresponds to a distinct NMR spectrum.

## E EXTRA CASE STUDIES OF IE-META-MMA

In molecular A in Figure E.1, atom 13 and atom 14 are tertiary carbons (attaching to 3 carbons and 1 hydrogen) and on the same 5-member ring, corresponding to the ppm of 34.3 and 35.6, respectively. The similar local content of these two atoms fools SP-ID and WP-ID. In addition, WP-ID fails with more atomic alignments. The molecular B is chemical symmetric regarding atom 0. Thus, atom 1 and atom 3 correspond to the same peak on the spectra. The ppm of atom 1 and atom 3 is 114.2, the ppm of atom 2 and atom 4 is 110.0. While there is 4.2 difference, SP-ID and WP-ID fails to pick up right alignment for atom 1 and atom 3. In contrast, K-ID succeed to align the atoms with peaks in both molecules.

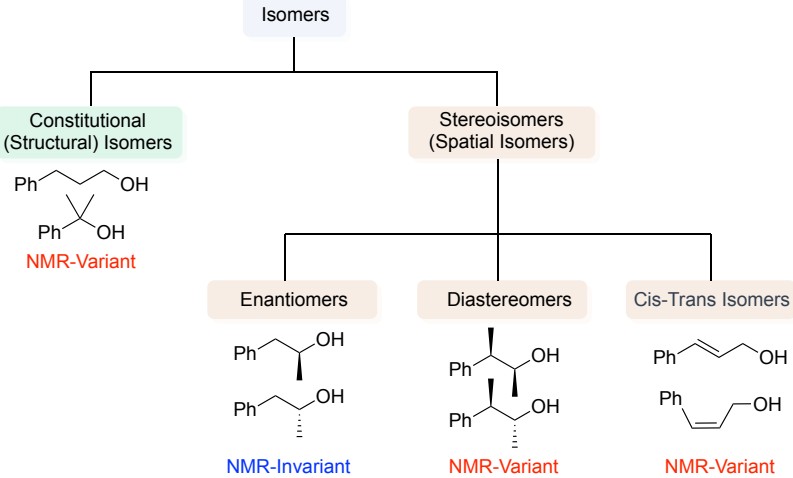

Figure D.1: Isomer categories

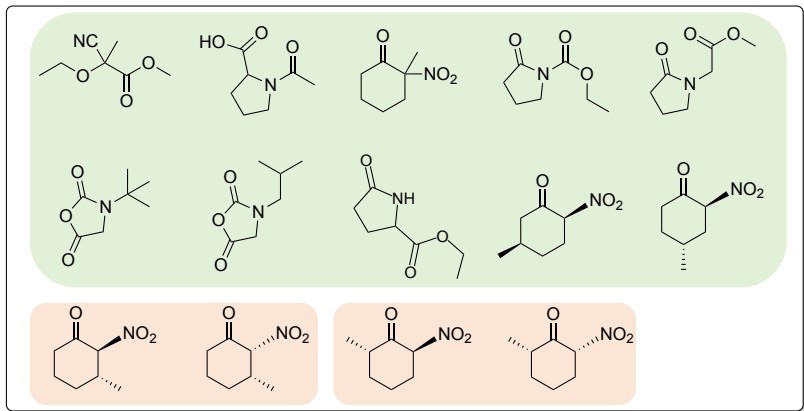

Figure D.2: Isomer demo for $C_7H_{11}NO_3$

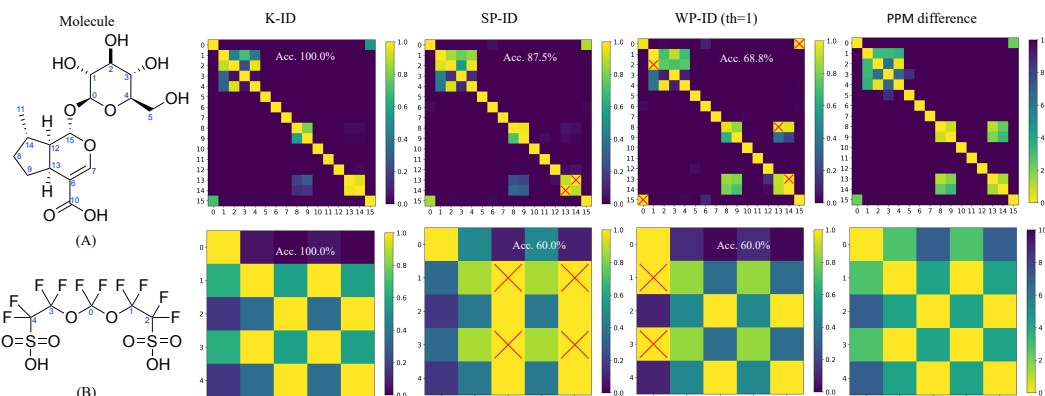

Figure E.1: Extra case studies of IE-Meta-MMA. Yellow cells in the PPM differerence represent the ground truth alignment, and red cross represents the wrong alignment.

