# OpenReview forum: "Multil-Level Multimodal Alignment with Knowledge-Guided Instance-Wise Discrimination"
_ICLR.cc/2024/Conference — ICLR 2024 Conference Withdrawn Submission_

### Official Review · Reviewer_svdX · 2023-10-28

**Soundness:** 2 fair
**Presentation:** 2 fair
**Contribution:** 2 fair
**Rating:** 3
**Confidence:** 3

**Summary:**

The paper introduces a new approach called K-M3AID for addressing the challenges of integrating meta-alignment into a multi-level multimodal alignment framework.
The paper highlights the importance of multimodal alignment in connecting contextually related information across different modalities.
The K-M3AID framework utilizes continuous knowledge variables with natural ordering for meta-alignment.
The paper discusses the three key components of the framework, i.e., the RS-MMA module, IE-Meta-MMA module, and Communication Channel, and presents its potential applications in chemistry, specifically in isomer recognition.
Empirical results demonstrate the effectiveness of K-M3AID in multiple zero-shot tasks.

**Strengths:**

1) The K-M3AID framework introduces a novel approach for integrating meta-alignment into a multi-level multimodal alignment framework, addressing the challenges of aligning both reducible substances and irreducible elements.
2) The paper demonstrates the potential applications of the K-M3AID framework in chemistry, particularly in solving challenging tasks such as isomer recognition. This highlights the practical relevance of the proposed approach.
3) Experimental results show the effectiveness of K-M3AID in zero-shot Spec2Mol task and Isomer Recognition task.

**Weaknesses:**

1. The paper's argumentation is unclear, and the motivation is not convincing. For instance:

	a. Why does IE-MMA, in conjunction with meta-learning, converge into multimodal meta-alignment?
	b. What specific levels are referred to by "multi-level"?

	c. The motivation for the integration of RS-MMA and IE-Meta-MMA lacks persuasiveness due to the unclear exposition of the respective advantages.

	d. How does MLMMA differ from multi-level RS-MMA?

	e. What does "meta-MMA" refer to?

	f. What is the nature of the dependence between RS-MMA and IE-Meta-MMA, and why is it considered a new challenge?

	g. While multi-modal alignment is a crucial topic in deep learning with applications across various multi-modal tasks, why does this paper primarily focus on chemistry? It initially conveys the impression of proposing a general multimodal alignment model but, in practice, concentrates solely on multimodal alignment within the field of chemistry. Additionally, the authors fail to provide sufficient justification for this specific focus.

2. In the related work section, the author lists relevant studies but lacks a comparative analysis of this work. There is a dearth of emphasis on what distinguishes or advantages this work.

3. Some explanations lack clarity, for instance, the explanation of why the "red cross" part results in "the wrong alignment" is inadequately addressed. A more comprehensive analysis should be presented, including instances where the proposed model may fail.

**Questions:**

see weakness.

---

### Official Review · Reviewer_XYBY · 2023-10-30

**Soundness:** 2 fair
**Presentation:** 2 fair
**Contribution:** 3 good
**Rating:** 5
**Confidence:** 3

**Summary:**

The paper propose a multimodal alignment method named K-M3AID to learn molecular and atomic representations. Using domain-specific features, the author introduce the knowledge span guided loss to facilitate contrastive learning. Model are trained on the nmrshiftdb2 dataset, with downstream zero-shot tasks to validate the effectiveness.

**Strengths:**

1. The paper proposes an interesting approach to combine molecular's natural ordering to contrastive representation learning.
2. The presentation of model structure and case study is clear and explicit.

**Weaknesses:**

1. The authors should provide more baseline results of the tasks to demonstrate the effectiveness of the proposed method.
2. The authors fail to provide a detailed explanation of the principles behind the Communication Channel, nor how it was implemented. Furthermore, there is a lack of ablation studies for this module.
3. There are factual errors in the paper:
    1. In equation 2, the CL_RS(i) loss is missing a negative sign.
    2. In Appendix D.1 enantiomers are NMR-invariant not NMR-variant.
4. The author should enhance their writing, many grammar mistakes and typos are spotted in the paper:
    1. "Multil-Level" in the title.
    2. "and CL_IE the contrastive" in section 3.2.
    3. "Appendix A As a result" in section 3.3.
    4. "the molecules were collected over 1 million from Pub-Chem" in section 4.1.1.

**Questions:**

1. In section 4.3, what is the explicit loss function of SP-ID and WP-ID given the Strong Pair and Weak Pair in formula 12 and 13?
2. When you prepare the data for Spec2Mol retrieval task, how to align spectra from nmrshiftdbs to molecules from Pub-Chem dataset?
3. What is the difference between the task of molecular alignment and Spec2Mol retrieval?
4. In section 4.1.2, there seems to be a gap between a=0.2 and a=0.5. Why do you claim the optimal setting is 0.2 without testing parameters between 0.2 and 0.5?
5. Do you envision broader applications for the method you've proposed in the field of molecular chemistry? Additionally, in which other domains do you believe it could be applied?
6. In the isomer recognition task, you have only provided 10 case studies, all with 100% accuracy. Is it possible to test the model on a larger dataset and more challenging tasks?

---

### Official Review · Reviewer_VTNJ · 2023-11-01

**Soundness:** 3 good
**Presentation:** 3 good
**Contribution:** 2 fair
**Rating:** 5
**Confidence:** 2

**Summary:**

This paper proposes a novel approach called K-M3AID (Knowledge-guided Multi-level Multimodal Alignment with Instance-wise Discrimination) for multimodal alignment, which involves the operation of both reducible substances and irreducible elements. K-M3AID utilizes continuous knowledge variables for meta-alignment and instance-wise discrimination, which expands the scope of contrastive learning from confined comparisons to unrestricted comparisons, thus mitigating the potential introduction of human bias. Empirical studies conducted on complex molecular structures demonstrate the effectiveness and reliability of K-M3AID.

**Strengths:**

1. This K-M3AID approach is straightforward but performs good zero-shot performance. It is interesting to incorporate the knowledge span in multimodal alignment.
2. The paper is well-organized and easy to follow. The clarity of the paper is also noteworthy.

**Weaknesses:**

1. It is beneficial to include more diverse datasets to demonstrate the effectiveness of K-M3AID across a range of molecular contexts.
2. While the paper does provide a novel approach to multimodal alignment, it would be beneficial to provide a more detailed comparison with existing approaches to demonstrate the superiority of K-M3AID.
3. More ablation analysis is needed in my opinion. It is better to understand the role between the RS-MMA module and the IE-Meta-MMA Module.

**Questions:**

Why does only the pre-trained spectrum encoder remain fixed in Section 3.1?